# Serum Visfatin/NAMPT as a Potential Risk Predictor for Malignancy of Adrenal Tumors

**DOI:** 10.3390/jcm11195563

**Published:** 2022-09-22

**Authors:** Nadia Sawicka-Gutaj, Hanna Komarowska, Dawid Gruszczyński, Aleksandra Derwich, Anna Klimont, Marek Ruchała

**Affiliations:** Department of Endocrinology, Metabolism and Internal Medicine, Poznan University of Medical Sciences, 60-355 Poznan, Poland

**Keywords:** nicotinamide phosphoribosyltransferase, NAMPT, visfatin, adrenocortical carcinoma, benign adrenocortical tumor

## Abstract

Adrenocortical carcinomas (ACC) are rare endocrine malignancies, often with a poor prognosis. Visfatin/NAMPT regulates a variety of signaling pathway components, and its overexpression has been found in carcinogenesis. Our study aimed to assess the clinical usefulness of visfatin/NAMPT serum level in discriminating between ACC and benign adrenocortical tumors. Twenty-two patients with ACC and twenty-six patients with benign adrenocortical tumors were recruited. Fasting blood samples were collected from each patient, and visfatin serum levels were measured with the ELISA Kit. Clinical stage, tumor size, Ki67 proliferation index, hormonal secretion pattern, and follow-up were determined in ACC patients. Patients with ACC had significantly higher visfatin serum concentrations (7.81 ± 2.25 vs. 6.08 ± 1.32 ng/mL, *p*-value = 0.003). The most advanced clinical stage with metastases was associated with significantly elevated visfatin levels (*p*-value = 0.022). Based on ROC analysis, visfatin serum concentrations higher than 8.05 ng/mL could discriminate ACC with a sensitivity of 50.0% and specificity of 92.3%. Univariate Cox regression indicated that tumor size was significantly related to shorter survival, and the visfatin level was borderline significant in all patients (HR = 1.013, *p*-value = 0.002, HR = 1.321, *p*-value = 0.058). In the Kaplan-Meier method, patients with visfatin serum concentrations higher than 6.3 ng/mL presented significantly lower survival probability (*p*-value = 0.006). Serum visfatin/NAMPT could be a potential risk predictor for the malignancy of adrenal tumors. However, further studies are needed on this subject.

## 1. Introduction

Adrenal tumors are common neoplasms, accounting for 3–10% of tumors in the adult population [1,2]. In most cases, they are diagnosed incidentally during imaging performed for other reasons than adrenal disease. As a result of the improvement of radiological techniques and their common use, the discovery of adrenal incidentalomas, defined as lesions greater than 1 cm, is still increasing, ranging from 0.5% in children to 10% in elderly patients in computed tomography (CT) scans [3]. Adrenal tumors may originate either from the cortex or medulla or be classified as secondary lesions. They can be categorized as benign or malignant, functional (hormone secreting), which can lead to clinical conditions such as Cushing’s syndrome or hyperaldosteronism, or non-functional [1,2,3,4,5,6]. The most frequent finding among incidentalomas is non-functional benign adenomas, comprising 80% of cases. Albeit uncommon, aggressive adrenocortical carcinoma (ACC) is a malignant tumor, ranging at 1.2%–11% of incidentaloma cases [4], with an estimated annual incidence of 0.7–2 cases/year and a worldwide prevalence of 4–12 cases per million/year [7]. The age of distribution is bimodal, with peaks in the first and fourth/fifth decade, with female predominance [1,2,6]. It is mostly sporadic but may occur as a component of genetic syndromes such as Li-Fraumeni syndrome, Beckwith-Wiedemann syndrome, or multiple endocrine neoplasia type I (MEN 1). Among the patients with ACC, approximately 50% are asymptomatic or present with symptoms related to mechanical effects of tumor growth. The other half of patients with functional tumors may present with Cushing’s syndrome, virilization, and rarely feminization or hyperaldosteronism [5,6,7]. The adrenal surgery for ACC should include a complete *en bloc* excision with resection of oligometastatic disease. In addition, adjuvant mitotane therapy (such as monotherapy or combined with chemotherapy) is recommended in case of ACC with advanced progression or a higher risk of recurrence [8]. In contrast, surgical intervention for adrenal incidentalomas is indicated only in patients with autonomous cortisol secretion and depends on accompanying factors such as the likelihood of malignancy, degree of hormone excess, age, general health, and patient preference. Asymptomatic, non-functioning, unilateral adrenal masses with obvious benign radiological features are not recommended for surgical treatment [9].

Overweight or obesity are common conditions accompanying hormonally active as non-functionating adrenal tumors. Adipose tissue works as an endocrine gland secreting adipocytokines such as resistin, leptin, adiponectin, and visfatin, affecting insulin resistance and metabolic syndrome [10,11]. There are not many studies specifying the role and contribution of adipocytokines in the biology of ACC [10,12,13]. Visfatin, also known as nicotinamide phosphoribosyltransferase (NAMPT) or pre-B-cell-enhancing factor, is a novel adipocytokine expressed mostly in visceral adipose tissue, though detectable in most cell types such as macrophages, leucocytes, primary glial cells, or fibroblasts. Its increased concentration in serum is associated with the development of obesity, diabetes mellitus 2, carcinogenesis, and other inflammatory disorders. It can act as an enzyme, cytokine, or growth factor and regulate a variety of signaling pathway components [14]. Visfatin/NAMPT stimulates the production of many proinflammatory cytokines and enhances the production of IL-1α, IL-1β, ΙL-6, IL-8, and TNF-α [14,15]. Previous research shows a correlation between increased level of serum visfatin and malignant potential, stage progression, and prognosis [16,17]. *NAMPT* overexpression has been found in various types of malignancy, such as acute-type adult T-cell leukemia/lymphoma cells and breast, thyroid, endometrial, bladder, pancreatic, colorectal, gastric, and prostate cancers [14,15,18,19].

So far, to our best knowledge, this is the first research studying the role of visfatin in ACC pathophysiology. Our study aimed to assess the clinical usefulness of visfatin/NAMPT serum level in discriminating between ACC and benign adrenocortical tumors.

## 2. Materials and Methods

### 2.1. Study Design and Patient Enrollment

This was an observational study with consecutive enrollment of patients adrenalectomized from 2013 to 2017. Indications for adrenalectomy included suspicion of malignancy, regrowth of tumor mass, and hormonal activity of the tumor (pheochromocytoma, Cushing’s syndrome, Conn’s syndrome). All patients signed informed consent.

Histopathological diagnosis of ACC was based on Weiss criteria [20]. Tumor size, tumor stage at diagnosis according to the European Network for the Study of Adrenal Tumors classification [9], Ki67 proliferation index, hormonal secretion pattern, and follow-up were evaluated in ACC patients.

The study was conducted according to the guidelines of the Declaration of Helsinki and was approved by the Bioethics Committee of Poznan University of Medical Sciences [21].

### 2.2. Laboratory Analysis

Fasting blood samples were collected from each patient. Visfatin/NAMPT serum level was assessed with an ELISA Kit (Phoenix Pharmaceuticals Inc., Burlingame, CA, USA). Total cholesterol (TC), low-density lipoprotein (LDL), high-density lipoprotein (HDL), and triglyceride (TAG) levels were measured using a spectrophotometric method.

### 2.3. Statistical Analysis

Descriptive statistics of quantitative variables were presented in the form of means and standard deviations (if the normal distribution) or medians and quartile ranges (if non-compliant with the normal distribution). The data comparison between the two groups was performed with parametric tests (for the normal distribution), the un-paired Student’s *t*-test (for equal variances), Welch’s *t*-test (for unequal variances), and the non-parametric *U* Mann-Whitney’s test (for non-compliance with the normal distribution). Compliance with the normal distribution was assessed with the Shapiro-Wilk test and the equality of variances with Levene’s test. Qualitative variables were compared with Pearson’s Chi-square test.

Due to the compliance with the normal distribution, Pearson’s correlation coefficients were determined for visfatin serum concentration and other clinical parameters. The multiple linear regression model was developed to predict visfatin levels in ACC patients.

The receiver operating characteristics (ROC) analysis was performed to assess the value of the visfatin serum level to discriminate the patients with ACC and benign adrenocortical tumors. The potential cut-off point was proposed based on Youden’s index. In addition, the univariate logistic regression model incorporating visfatin serum level as a discriminating predictor for ACC was conducted with V-fold cross-validation and the Hosmer–Lemeshow test.

In addition, the survival analysis using the Cox regression and the Kaplan-Meier method was performed. For the second method, the cut-off point for division into subgroups was determined using the Weight of Evidence analysis. Survival curves were compared with the log-rank test.

The significance level was set at α = 0.05 for all analyses. The statistical analysis was performed with Statistica 13.3 (Statsoft, Cracow, Poland). The sample size and power analyses were conducted with MedCalc 19.5.3 (MedCalc Software Ltd., Ostend, Belgium) and RStudio 1.2.5033 (RStudio Inc., Boston, MA, USA).

## 3. Results

### 3.1. Clinicopathological Characteristics

A total of twenty-two patients with adrenocortical carcinomas (12 females, 10 males, mean age 51.3 ± 12.4) and twenty-six patients with benign adrenocortical tumors (13 females, 13 males, mean age 55.5 ± 12.6) were recruited. Patients with benign adrenocortical tumors included twenty with adenomas, three with myelolipomas, two with nodular hyperplasia, and one adrenal cyst. Table 1 presents clinical characteristics of both groups.

Individual data for patients with ACC are presented in Table 2.

### 3.2. Visfatin Serum Concentrations

Patients with adrenocortical carcinomas had significantly higher visfatin serum concentrations compared to patients with benign adrenocortical tumors—respectively, 7.81 ± 2.25 vs. 6.08 ± 1.32 ng/mL, *p*-value = 0.003 for the Welch’s *t*-test (Figure 1). Power analysis was determined at the level of 84.0% (Appendix A).

Visfatin serum concentrations did not differ between males and females in both groups with ACC and benign adrenocortical tumors. In contrast, patients demonstrating the most advanced clinical stage with metastases had significantly elevated visfatin levels in comparison to the rest of the patients—respectively, median with quartile ranges 8.1 (6.8–10.2) vs. 6.0 (5.6–7.0) ng/mL, *p*-value = 0.022 for the *U* Mann-Whitney’s test (non-compliance with the normal distribution).

No significant correlations between clinicopathological parameters and visfatin concentrations were observed in either group with ACC and benign adrenocortical tumors (Table 3). However, by constructing a model of multiple linear regression consisting of age and size of tumor for patients with ACC, both predictors achieved statistical significance, and the model had the adjusted R^2^ coefficient of determination 34.5% (Table 4). Thus, younger patient age and smaller malignant tumor sizes were associated with higher serum concentrations of visfatin. A decrease in age of one year and tumor size of one millimeter was accompanied by an increase in visfatin serum levels of about 0.1 and 0.015 ng/mL, respectively.

In addition, ROC analysis detected visfatin serum concentrations higher than 8.05 ng/mL as a biomarker of ACC with a sensitivity of 50.0% and specificity of 92.3% (Figure 2). Power analysis was determined at the level of 85.0% (Appendix A). In addition, the univariate logistic regression model indicated visfatin serum concentration as a significant predictor for ACC, with the odds ratio equal to 1.738 (Table 5). The results of the V-fold cross-validation showed the good quality of this model (Appendix A). The Hosmer-Lemeshow confirmed the goodness of fit (*p*-value > 0.05).

### 3.3. Survival Analysis

Based on the univariate Cox regression, tumor size was significantly associated with shorter survival, and visfatin level was borderline significant in the whole group (Table 6). However, no significant predictors of mortality risk were observed for patients with ACC. Thus, visfatin serum concentrations could not determine to predispose to a shorter survival time significantly.

The Kaplan-Meier method was performed for all patients with adrenocortical tumors by dividing them into two subgroups at visfatin level 6.3 ng/mL (as determined by the Weight of Evidence analysis). Patients with higher visfatin serum concentrations demonstrated significantly lower survival probability (Figure 3).

## 4. Discussion

In our study, we observed that visfatin serum concentrations were higher in patients with adrenocortical carcinomas compared to patients with benign adrenocortical tumors. For both subgroups, no correlation was found between visfatin levels and tumor size. In addition, visfatin levels were not correlated with age and BMI, regardless of subgroup classification. However, according to the multiple linear regression, higher visfatin serum concentrations seemed to be related to the smaller tumor size and younger age in ACC patients.

Based on the ROC analysis, we demonstrated the potential value of visfatin to discriminate patients with adrenocortical carcinomas from those with benign adrenocortical tumors. Moreover, patients with adrenocortical tumors presenting higher visfatin levels had a significantly lower survival probability.

Since adrenal cancers are relatively rare, a small study group was assembled, which may lead to a lack of statistical significance for some of the results. However, the obtained differences in visfatin levels between ACC and benign adrenocortical tumors demonstrated relatively high statistical power. Nevertheless, it should be emphasized that, to our knowledge, this is the first study to evaluate the relationship between adrenal tumor progression and visfatin levels as a mediator of proinflammatory processes. In this study, we correlated visfatin concentrations with clinicopathological parameters. The study limitations should also include a relative heterogeneity of histopathological diagnosis in patients with benign adrenocortical tumors, as well as an observational study design that does not allow for assessing cause–effect relationships.

In another study, Atasoy et al. [22] observed significantly higher visfatin concentrations in patients with hormonally inactive adrenal adenoma compared to healthy controls. An analogous relationship was demonstrated after the adjustment for age, BMI, the presence of diabetes mellitus, and hypertension. The mean concentration of visfatin in patients with adrenal adenomas was 4.40 ± 3.05 ng/mL, which was close to the levels of visfatin determined for the benign adrenocortical tumors in this study. In addition, there was no significant difference between the groups in terms of mean fasting TG, LDL-C, HDL-C, and total cholesterol levels. Thus, the authors suggest that visfatin may be associated with adrenal adenoma pathophysiology. Similarly, we found no correlation between lipid profile parameters and visfatin levels in both groups with ACC and benign adrenocortical tumors.

On the other hand, Akkus et al. [10] assessed serum adipocytokine levels, including visfatin, leptin, resistin, omentin 1, and adiponectin in patients with non-functional adrenal incidentalomas (NFAI). These patients demonstrated significantly elevated leptin and resistin levels, as well as significantly decreased adiponectin levels. The comparison of visfatin and omentin 1 concentrations was non-significant. Additionally, there was no significant correlation between adipocytokines levels and adenoma size. The initial diagnosis based on the computed tomography (CT) scan, despite the possibility of assessing the adrenal tumor imaging phenotype, may be inconclusive, and the final diagnosis requires histopathological examination [23]. Therefore, increased visfatin levels could be a potential predictor of tumor malignancy.

In addition, previous studies have shown that visfatin/NAMPT concentrations are increased in several malignancies, such as colorectal carcinoma, gastric cancer, hepatocellular carcinoma, bladder cancer, breast cancer, endometrial cancer, and other carcinomas [16,24,25,26,27,28]. Based on a meta-analysis by Mohammadi et al. [29], higher levels of visfatin are associated with a higher risk of developing cancer. In the literature, visfatin/NAMPT elevation also shows an association with the presence of metastases, aggressive tumor stage and poor clinical prognosis [30]. In our study, ACC patients in stage IV with metastases showed a tendency to higher serum visfatin values. Previously, in thyroid cancer patients, we observed the overexpression of NAMPT in tissue samples, which was related to more advanced tumor stages with metastases to the lymph nodes [15]. However, based on our recent study, the serum visfatin/NAMPT levels could not be found as a potential marker for the detection of papillary thyroid cancer [31].

Although mentioned studies report the overexpression of visfatin/NAMPT in oncological patients, the source of this proinflammatory protein is still unclear. Visfatin/NAMPT is thought to be primarily secreted by adipose tissue, although neoplastic tumors are also considered as additional sources of increased levels of visfatin/NAMPT in this group of patients [32]. In previous studies, elevated levels of visfatin/NAMPT have been found in various tumor tissues, such as thyroid malignancies, ovarian serous adenocarcinomas, and malignant lymphomas [15,33,34]. Moreover, *in vitro* studies showed the increased secretion of visfatin/NAMPT from inter alia, melanoma cells, or human pancreatic adenocarcinoma cells [35,36]. Present studies in adrenal tumors may suggest that visfatin/NAMPT originates from tumors themselves due to no confirmed correlations with the lipid profiles of patients.

From the point of view of pathophysiology, visfatin/NAMPT is believed to demonstrate multidirectional mechanisms of action. Among the most important are inducing the production of inflammatory cytokines and increasing the activity of antioxidative enzymes. In the apoptotic state, visfatin-induced antioxidative activity results in increased viability of the cancer cells. Another mechanism that enhances the cancer cells survival is participation in the NAD generation pathway. Furthermore, the increased expression of NAMPT by influencing Sirt-1, vascular endothelial growth factor (VEGF), and matrix metalloproteinase (MMP) activity significantly stimulates angiogenesis and thus contributes to the progression of tumor development [37,38].

As a key regulator of cancer cell metabolism, visfatin/NAMPT is elevated in oncological patients and may thus be a therapeutic target. Cancer cells are more susceptible to NAMPT inhibitors compared to normal cells [32]. The classic NAMPT inhibitors include FK866 or APO866, CHS-828 (also known as GMX1778), and its pro-drug (GMX1777). Additionally, combining these specific inhibitors with other antineoplastic agents or radiotherapy may help to effectively overcome multidrug resistance in cancer treatment [39,40,41]. For instance, GMX1778 seems to be a potential radiosensitizer in combination with 177Lu-DOTATATE for the treatment of neuroendocrine tumors. In GOT1-bearing mice, prolonged antitumor response was observed for this combined therapy [42]. Moreover, *NAMPT mRNA* expression level was found as a predictor for the sensitivity of cells to FK866. The addition of FK866 could enhance the in vitro efficacy of gemcitabine in the pancreatic ductal adenocarcinoma subpopulation [43]. To the best of our knowledge, there are no reports available on the effectiveness of the use of NAMPT inhibitors in the treatment of adrenal malignancies.

## 5. Conclusions

Serum visfatin/NAMPT could be a potential risk predictor for the malignancy of adrenal tumors. However, further studies are needed to explain the clinical role of visfatin in ACC pathophysiology.

## Figures and Tables

**Figure 1 jcm-11-05563-f001:**
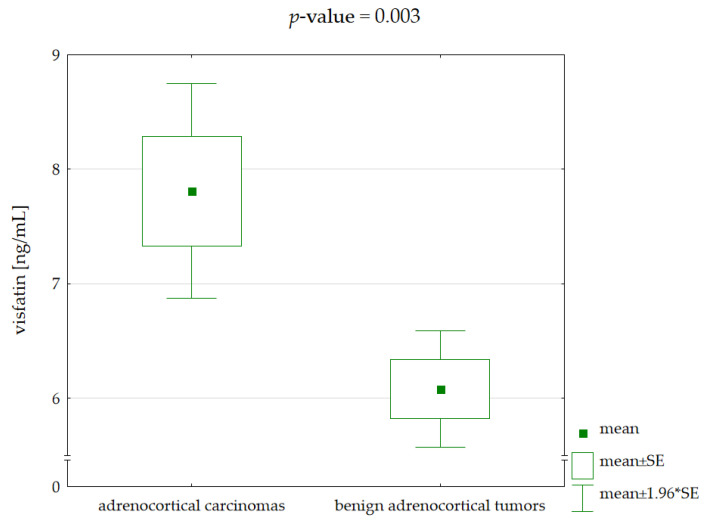
Comparison of visfatin serum concentrations between patients with adrenocortical carcinomas and benign adrenocortical tumors (*p*-value for Welch’s *t*-test).

**Figure 2 jcm-11-05563-f002:**
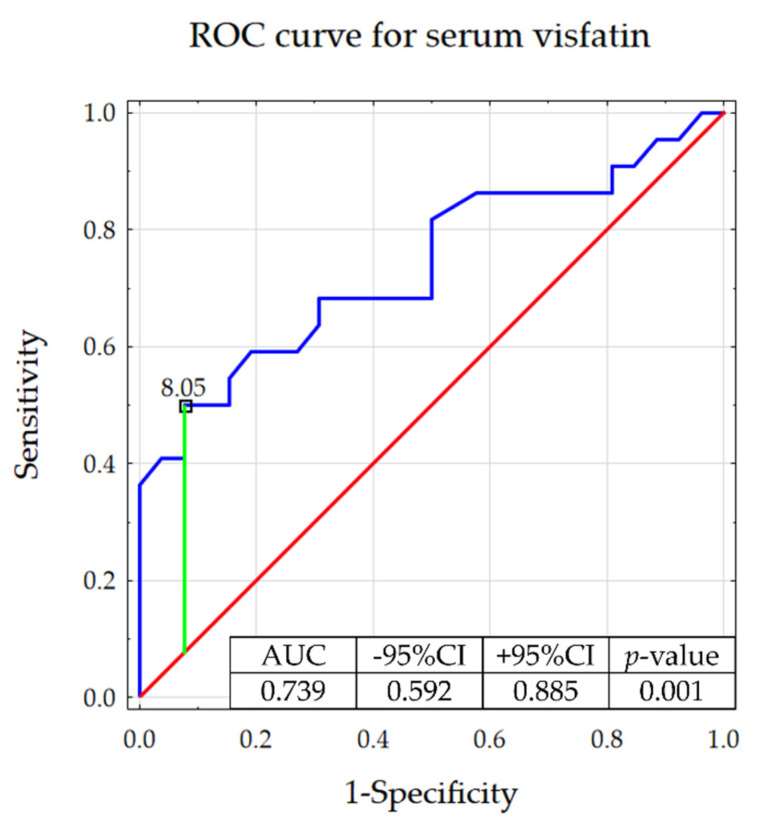
Receiver operating characteristic curve determining the potential of serum visfatin to discriminate between adrenocortical carcinomas and benign adrenocortical tumors (with presented proposed cut-off based on the Youden’s index).

**Figure 3 jcm-11-05563-f003:**
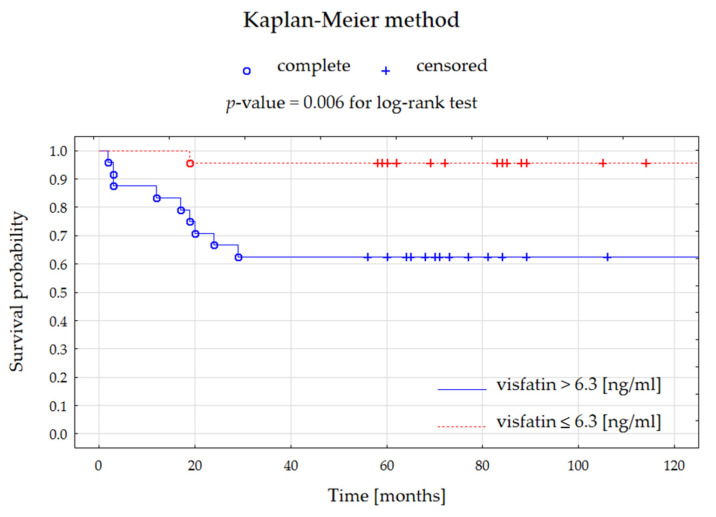
Kaplan-Meier curves presenting the survival probability for patients with all adrenocortical tumors depending on visfatin serum level for 10 years of follow-up.

**Table 1 jcm-11-05563-t001:** Comparison of clinical characteristics of patients with adrenocortical carcinomas and benign adrenocortical tumors (data presented as mean ± standard deviation).

	Adrenocortical Carcinomas *n* = 22	Benign Adrenocortical Tumors *n* = 26	*p*-Value
Age, years	51.3 ± 12.4	55.5 ± 12.6	0.256 ^
Gender (females), *n*	12	13	0.753 ^^^
BMI, kg/m^2^	26.4 ± 4.0	27.9 ± 5.1	0.358 ^
TC, mg/dL	186.9 ± 45.4	196.7 ± 47.2	0.537 ^
HDL, mg/dL	56.4 ± 25.1	53.4 ± 14.5	0.688 ^^
LDL, mg/dL	104.6 ± 40.0	114.3 ± 35.5	0.480 ^
TAG, mg/dL	116.6 ± 54.8	150.2 ± 37.0	0.056 *^^
Tumor size, mm	136.4 ± 62.3	41.7 ± 23.4	<0.001 *^^
Metastases, *n*	13	0	<0.001 *^^^
Deceased, *n*	12	0	<0.001 *^^^
Survival median, months	42.6	-	-

Legend: *, statistically significant difference; ^, for the unpaired Student’s *t*-test; ^^, for the Welch’s *t*-test; ^^^, for the Pearson’s Chi-square test; BMI, body mass index; TC, total cholesterol; LDL, low-density lipoprotein; HDL, high-density lipoprotein; TAG, triglycerides.

**Table 2 jcm-11-05563-t002:** Detailed clinical characteristics of patients with adrenocortical carcinomas.

	Age (Years)	Gender	Clinical Stage	Tumor Size (mm)	Ki-67 (%)	Hormonal Activity	Follow-up (Months)	Deceased
ACC1	26	F	II	70	8	n/a	73	−
ACC2	41	F	II	210	1	n/a	68	−
ACC3	42	F	II	110	5	n/a	146	−
ACC4	48	F	II	96	NR	A4	114	−
ACC5	57	F	II	60	1	DHEAS	77	−
ACC6	71	F	II	100	25	n/a	59	−
ACC7	68	M	II	63	1	n/a	19	+
ACC8	41	F	III	230	80	n/a	58	−
ACC9	58	F	III	200	15	n/a	167	−
ACC10	43	F	IV	62	35	n/a	128	−
ACC11	63	F	IV	170	30	n/a	29	+
ACC12	63	F	IV	80	NR	n/a	2	+
ACC13	69	F	IV	130	NR	TST	NR	+
ACC14	38	M	IV	NR	NR	n/a	329	+
ACC15	40	M	IV	130	40	DHEAS	24	+
ACC16	40	M	IV	192	NR	DHEAS	3	+
ACC17	42	M	IV	150	50	DHEAS	12	+
ACC18	46	M	IV	260	NR	n/a	19	+
ACC19	52	M	IV	200	15	DHEAS A4	17	+
ACC20	56	M	IV	170	25	TST	20	+
ACC21	58	M	IV	125	20	n/a	71	−
ACC22	67	M	IV	57	30	TST	3	+

Legend: A4, androstenedione; DHEAS, dehydroepiandrosterone sulphate; TST, testosterone; F, female; M, male; NR, not reported; n/a, not applicable.

**Table 3 jcm-11-05563-t003:** Correlation coefficients (R) for visfatin serum level and clinicopathological parameters.

	Adrenocortical Carcinomas		Benign Adrenocortical Tumors	
	R_P_	*p*-Value	R_P_	*p*-Value
Age, years	−0.349	0.112	−0.166	0.426
BMI, kg/m^2^	0.015	0.961	0.095	0.650
TC, mg/dL	−0.140	0.634	0.012	0.958
HDL, mg/dL	0.029	0.922	0.352	0.108
LDL, mg/dL	−0.082	0.790	−0.284	0.213
TAG, mg/dL	−0.288	0.319	−0.190	0.384
Tumor size, mm	−0.308	0.174	0.160	0.446

Legend: R_P_, Pearson’s correlation coefficients (in compliance with the normal distribution); BMI, body mass index; TC, total cholesterol; LDL, low-density lipoprotein; HDL, high-density lipoprotein; TAG, triglycerides.

**Table 4 jcm-11-05563-t004:** Parameters of predictors incorporated into the multiple linear regression model for visfatin serum concentrations in patients with adrenocortical carcinomas.

	Standardized β	SE	*p*-Value
Intercept			<0.001 *
Age, years	−0.580	0.187	0.006 *
Tumor size, mm	−0.454	0.187	0.026 *

Legend: SE, standard error; *, statistically significant predictor.

**Table 5 jcm-11-05563-t005:** Parameters of visfatin serum level incorporated into the univariate logistic regression model.

	β	SE	Wald Stat.	*p*-Value	Odds Ratio	Confidence OR −95%	Confidence OR 95%
Intercept	−3.964	1.418	7.813	0.005 *	0.019	0.001	0.306
Visfatin, ng/mL	0.553	0.203	7.412	0.006 *	1.738	1.168	2.588

Legend: SE, standard error; OR, odds ratio; *, statistical significance <0.05.

**Table 6 jcm-11-05563-t006:** Hazard ratios for predictors of mortality risk in the univariate Cox regression.

	Adrenocortical Carcinomas				All Patients			
	HR	−95%CI	+95%CI	*p*-Value	HR	−95%CI	+95%CI	*p*-Value
Age, years	1.031	0.978	1.088	0.259	1.004	0.954	1.056	0.887
BMI, kg/m^2^	1.142	0.925	1.408	0.218	1.033	0.879	1.215	0.692
Visfatin, ng/mL	1.014	0.764	1.345	0.925	1.321	0.990	1.763	0.058 ^
Tumor size, mm	1.003	0.993	1.013	0.587	1.013	1.005	1.021	0.002 *

Legend: BMI, body mass index; HR, hazard ratio; CI, confidence interval; *, statistically significant predictor; ^, borderline significant predictor.

## Data Availability

The datasets generated and/or analyzed during the current study are available from the corresponding author on reasonable request.

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
