# Peer review of "Serum Visfatin/NAMPT as a Potential Risk Predictor for Malignancy of Adrenal Tumors"

_jcm, 2022, doi:10.3390/jcm11195563_

Round 1

Reviewer 1 Report

I have some difficulty with this study. The authors seem to suggest in the beginning the visfatin could play a role in oncogenesis, but the reason for this assumption is, to say the least, very vague and largely based on comparison to other tumors, where no further explanations are given. The obvious other explanation is that the visfatin levels correlate with tumor size and in several instances the authors tell us this is so. But then after a statistical ploy it proves that the younger patients with malignancies have lower visfatin levels. This is confusing and with the number of cases involved in this study it is questionable if such a subgroup evaluation is actually reliably possible, In any case what "younger patients"are is not stated.. The whole thing looks like a somewhat clumsy attempt to try to explain the correlation between visfatin levels and a poorer prognosis. To me the explanation that a correlation between tumor size and visfatin levels is far more likely. And to make this manuscript publishable this should be clearer presented and discussed.

In addition, it is also questionable whether the benign tumors are proper controls for this study. This group is very heterogenous and the diversity might effect the outcome of the comparison. It would be much better to compare the ACC with the adenomas only.

In itself the finding is of interest but the current presentation is not good enough

Author Response

We sincerely thank the Reviewer for constructive and valuable comments.

Our point-by-point responses to them are given below:

  1. The role of visfatin in oncogenesis is described in the context of other cancers, as this study describes adrenal cancer for the first time.
  2. In the primary analysis, visfatin levels were positively correlated with tumor size in the overall patient group; however, it was related to the fact that malignant tumors were larger than benign tumors, and in the group of malignant tumors this correlation was inverse. In summary, the highest visfatin secretion was observed in smaller malignancies, possibly related to their progression. In the revised version, the correlations for the whole group were removed, leaving for ACC and adding for benign tumors [Table 3].
  3. As for the doubts regarding the relationships of the visfatin level with the age and size of the tumor, they result from the interpretation of the results of multiple linear regression (with the standardized beta coefficients presented in Table 4). An increase in the value of the predictor by its standard deviation results in a change in the value of the explained variable by the product of the standard deviation of this variable with the standardized beta coefficient. On this basis, a comment was added in the description of the results showing the converted dependencies on how the visfatin level changes in patients with ACC with a decrease in age by 1 year and with a smaller tumor size by 1 mm [lines: 174-176].
  4. In order to increase the homogeneity of the study group, partially following the Reviewer's suggestion, we decided to remove pheochromocytomas, originated from the adrenal medulla [Table 1]. In turn, we left all the benign tumors, located in the adrenal cortex as the "control group". We made this decision based on the pathophysiological basis of the location and development of ACC, as well as simulations of the potential power of the performed statistical tests. The suggested limitation of the “controls” to patients with adenomas (only 16 versus 22 with ACC) would reduce the power of statistically derived relationships below the desired 80%. All statistical analyses were performed for the new study group, reaching the testing power above 80%. Also, we added the comment about the heterogeneity of this group in the paragraph on the study limitations [lines: 246-247].
  5. In connection with the aforementioned changes in the study design and the new statistical analyses, the abstract, the description of the results, the discussion and the supplementary materials were revised accordingly.

All changes are marked in the revised manuscript using the "Track Changes" function in Microsoft Word.

Reviewer 2 Report

In this original work the authors tackle the challenging issue of the differential diagnosis between ACC and benign adrenal tumors, which has been poorly explored so far and does not benefit from strong clinical recommendations. The study is focused on the potentially clinical usefulness of 13 visfatin/NAMPT serum level in this context.

Using an analytical framework, the authors describe the results achieved and provide a comprehensive overview of the issue.

The overall level of the paper is pretty good and innovative.

The quality of the tables and figures is adequate.

The first part of the manuscript (the introduction section) provides useful information for the readers. The background of the topic gives the reader a comprehensive roundup of the adrenal tumors (features and epidemiology). However, some details could be added regarding the differences according to pathology and therapy of ACC vs benign adrenal tumors.  

- The materials and methods section, as well as the results, are clear and well conducted.   

- The discussion is complete and well written. However the last paragraph "As a key regulator of cancer cell metabolism, visfatin/NAMPT is elevated in oncological patients and may thus be a therapeutic target. Cancer cells are more susceptible to 275 NAMPT inhibitors compared to normal cells [30]. The classic NAMPT inhibitors include 276 FK866 or APO866, CHS-828 (also known as GMX1778), and its pro-drug (GMX1777). Additionally, combining these specific inhibitors with other antineoplastic agents or radio therapy may help to effectively overcome multidrug resistance in cancer treatment [37–279 39]" could be better explained, adding more details about the studies cited (type of cancer cells, type of experiments...).  

- The conclusion is well stated

Author Response

We sincerely thank the Reviewer for constructive and valuable comments.

Our point-by-point responses to them are given below:

  1. We added the suggested comment on the differences in the therapy of ACC vs benign adrenal tumors [lines: 53-61].
  2. Also, we added more details about the use of classic NAMPT inhibitors in cancer treatment in the last paragraph of the Discussion [lines: 308-315].

All changes are marked in the revised manuscript using the "Track Changes" function in Microsoft Word.